# Structures of channelrhodopsin paralogs in peptidiscs explain their contrasting K$^+$ and Na$^+$ selectivities

Takefumi Morizumi [1,8], Kyumhyuk Kim [1,8], Hai Li [2,8], Elena G. Govorunova [2,8], Oleg A. Sineshchekov [2], Yumei Wang[2], Lei Zheng [2], Éva Bertalan [3], Ana-Nicoleta Bondar[4,5], Azam Askari [6], Leonid S. Brown [6], John L. Spudich [2] ✉ & Oliver P. Ernst [1,7] ✉

Kalium channelrhodopsin 1 from *Hyphochytrium catenoides* (*Hc*KCR1) is a light-gated channel used for optogenetic silencing of mammalian neurons. It selects K$^+$ over Na$^+$ in the absence of the canonical tetrameric K$^+$ selectivity filter found universally in voltage- and ligand-gated channels. The genome of *H. catenoides* also encodes a highly homologous cation channelrhodopsin (*Hc*CCR), a Na$^+$ channel with >100-fold larger Na$^+$ to K$^+$ permeability ratio. Here, we use cryo-electron microscopy to determine atomic structures of these two channels embedded in peptidiscs to elucidate structural foundations of their dramatically different cation selectivity. Together with structure-guided mutagenesis, we show that K$^+$ versus Na$^+$ selectivity is determined at two distinct sites on the putative ion conduction pathway: in a patch of critical residues in the intracellular segment (Leu69/Phe69, Ile73/Ser73 and Asp116) and within a cluster of aromatic residues in the extracellular segment (primarily, Trp102 and Tyr222). The two filters are on the opposite sides of the photoactive site involved in channel gating.

In optogenetics, expression of microbial rhodopsins enables bidirectional control of neuronal activity by light-induced changes of the membrane potential[1]. Such microbial rhodopsins are light-activated ion channels and pumps that contain retinal linked via a Schiff base (RSB) to the opsin protein moiety. Optogenetic activation of neurons can be achieved with a variety of cation-conducting channelrhodopsins (CCRs), whereas available tools for optogenetic neuronal inhibition have so far been suboptimal. Among the latter, ion-pumping rhodopsins transport at most one ion per absorbed photon into or out of the cell, resulting in small currents, and produce undesired side effects[2]. Anion-conducting channelrhodopsins generate larger currents but may induce neurotransmitter release in the axonal terminals[3]. Recently we have identified two channelrhodopsins (ChRs) in the protist *Hyphochytrium catenoides* that show a greater permeability for K$^+$ than Na$^+$, for which we named them "kalium channelrhodopsins" (KCRs)[4]. Homologous proteins with similar properties have subsequently been found in other organisms by us and others[5,6]. KCRs lack the tetrameric K$^+$ selectivity filter contributed by two or four channel subunits, as found universally in voltage- or ligand-gated K$^+$ channels[7,8], and therefore the KCRs employ a different mechanism of K$^+$ selection which is unknown. When expressed in mammalian excitable cells, KCRs hyperpolarize the membrane via K$^+$ efflux, similar to

[1]Department of Biochemistry, University of Toronto, Toronto, ON, Canada. [2]Department of Biochemistry & Molecular Biology, Center for Membrane Biology, The University of Texas Health Science Center at Houston McGovern Medical School, Houston, TX, USA. [3]Physikzentrum, RWTH-Aachen University, Aachen, Germany. [4]Faculty of Physics, University of Bucharest, Măgurele, Romania. [5]Institute of Computational Biomedicine (IAS-5/INM-9), Forschungszentrum Jülich, Jülich, Germany. [6]Department of Physics and Biophysics Interdepartmental Group, University of Guelph, Guelph, ON, Canada. [7]Department of Molecular Genetics, University of Toronto, Toronto, ON, Canada. [8]These authors contributed equally: Takefumi Morizumi, Kyumhyuk Kim, Hai Li, Elena G. Govorunova. ✉e-mail: john.l.spudich@uth.tmc.edu; oliver.ernst@utoronto.ca

what naturally occurs during the repolarization phase of action potentials, and inhibit activity of mouse cortical[4,6] and hippocampal neurons[9], and human induced pluripotent stem cell-derived atrial cardiomyocytes[6].

The closest relatives of KCRs among other ChRs are "bacteriorhodopsin(BR)-like cation channelrhodopsins" (BCCRs) from cryptophyte algae, none of which exhibits K$^+$ selectivity[10]. BCCRs and KCRs share some structural and functional properties with haloarchaeal proton-pumping rhodopsins, including the highly conserved DTD residue motif in transmembrane helix 3 (TM3)[11]. As discussed elsewhere[11], the alternative name "pump-like channelrhodopsins" suggested later[12] for this group of proteins is misleading because the DTD motif is not conserved in other classes of ion-pumping rhodopsins besides archaeal and fungal proton pumps. Homologs of *H. catenoides* KCRs from other protists form a compact branch on the overall phylogenetic tree of ChR sequences[4,6]. However, only some of these homologs are K$^+$ selective, whereas others are Na$^+$ or even solely H$^+$ channels.

In addition to *Hc*KCR1 and *Hc*KCR2, the genome of *H. catenoides* encodes a third ChR, the protein sequence of which shows 74% identity and 86% similarity with *Hc*KCR1 in the seven transmembrane helix (7TM) domain that is sufficient for channel activity (Supplementary Fig. 1)[13]. Remarkably, this third channel that we named *Hc*CCR is Na$^+$ selective, with a relative permeability ratio $P_K/P_{Na} > 100$-fold smaller than that of *Hc*KCR1[4]. *Hc*CCR is further characterized by a smaller relative permeability to protons compared to Na$^+$ than that of typical chlorophyte CCRs, and so is potentially useful as an optogenetic tool for neuronal activation that is less likely to produce undesirable acidification of the cytoplasm.

Elucidation of the structural foundations of K$^+$ and Na$^+$ selectivity in ChRs is not only an important goal of fundamental ion channel research but is also required for the engineering of better optogenetic tools. In that regard, *Hc*KCR1 is of particular interest as a highly efficient tool for optical neuronal silencing[4,6,9]. The existence of so closely related proteins as *Hc*KCR1 and *Hc*CCR with such dramatic difference in the K$^+$ to Na$^+$ permeability ratio provides a unique opportunity for structure-function analysis. Previously we have found that mutations of only three residues convert *Hc*CCR to a K$^+$ selective channel with a $P_K/P_{Na}$ ratio of ~8[5]. Several other residues required for K$^+$ selectivity of *Hc*KCR1 are conserved in *Hc*CCR but are not sufficient to render it K$^+$ selective.

To gain mechanistic insight into ion selectivity, we present high-resolution cryo-electron microscopy (cryo-EM) structures of *Hc*KCR1 and *Hc*CCR trimers embedded in peptidiscs. We show that the native-like environment provided by the peptidisc offers an efficient alternative for obtaining high-resolution cryo-EM structures of ChRs, which so far have been available only by using nanodiscs[14–17]. The peptidisc reconstitution method relies on short amphipathic helical peptides without supplemented lipids, in contrast to reconstitution into nanodiscs, which requires selection of an ApoA1-derived nanodisc scaffolding protein of appropriate length and addition of precise amounts of matching lipids[18,19]. In peptidiscs, the so-called peptidisc peptide is based on the reverse ApoA1 sequence to form two 18-amino acid amphipathic helical peptide repeats separated by a proline residue. This peptide wraps around the purified target membrane protein with its bound annular lipids to replace detergent. The structural differences of *Hc*KCR1 and *Hc*CCR trimers together with electrophysiological analyses of mutants afford mechanistic insight into ion selectivity of these ChRs and provide the basis for tailoring new optogenetic tools.

## Results
### ChR trimers embedded into peptidiscs
The 7TM domains (amino acid residues 1–265) of *Hc*KCR1 and *Hc*CCR expressed in *Pichia pastoris* and solubilized in dodecylmaltoside

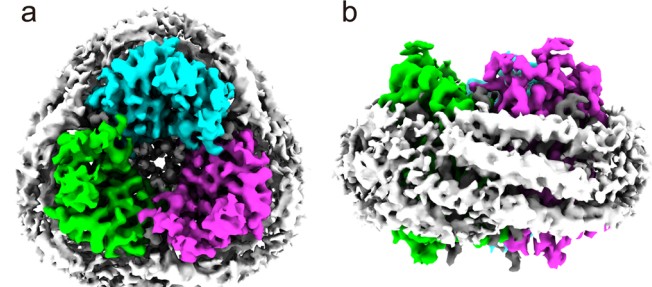

**Fig. 1 | Cryo-EM density map of *Hc*KCR1 embedded in peptidisc. a** Trimer of *Hc*KCR1 with C3 symmetry containing lipid molecules (grey) in the center (view from cytoplasm). The trimer is surrounded by few sterol and phospholipids and 37-residue peptidisc peptides. **b** Ring of helical peptidisc peptides encircling the *Hc*KCR1 trimer (cytoplasmic side up).

detergent showed single narrow peaks by size exclusion chromatography and typical rhodopsin absorption spectra (Supplementary Fig. 2). We reconstituted purified *Hc*KCR1 as well as *Hc*CCR at pH 7.5 into peptidiscs and imaged them by cryo-EM. The peptidisc environment had little effect on the function as determined by measuring the photochemical conversions of *Hc*KCR1 in liposomes and peptidiscs (Supplementary Fig. 3). The density maps of the ChR trimers obtained from cryo-EM imaging are shown for *Hc*KCR1 in Fig. 1. *Hc*KCR1 as well as *Hc*CCR show density maps of discs which are not round in shape as in the case of nanodiscs with embedded ChRs[14–17], but appear more three-cornered. This is due to the lower amount of lipids between the ChRs and the encircling scaffolding peptides or proteins, respectively. The nanodisc scaffolding proteins typically show two long parallel helical belts in the nanodisc plane surrounding the lipid disc. For the *H. catenoides* ChRs (*Hc*ChRs) in the present study, the shorter peptidisc peptides arrange in three loosely ordered helical stretches that run parallel at a small angle relative to the disc plane.

Cryo-EM single particle analysis of peptidisc-embedded *Hc*ChRs allowed us to obtain well defined density maps and to determine dark-adapted high-resolution structures at 2.88 Å for *Hc*KCR1 and 2.84 Å for *Hc*CCR with the resolution in the TM domain reaching 2.4 Å (Fig. 2, Table 1, Supplementary Figs. 4–6). The two ChR models are highly comparable and comprise in both cases residues 17 to 256 for the seven TM helices linked by three intracellular and three extracellular loops, as well as short N- and C-terminal regions (Fig. 2, Supplementary Fig. 1). Remarkably, the overall root-mean-square deviation (RMSD) between *Hc*KCR1 and *Hc*CCR structures is 0.450 Å and they lack any major conformational differences. Therefore, the ion selectivity differences of these two *Hc*ChRs must be due to subtle changes in local sites.

*Hc*KCR1 and *Hc*CCR in peptidiscs exhibit trimeric assembly (Figs. 1 and 2), characteristic of haloarchaeal H$^+$ pumps[20] and also found in nanodisc-reconstituted *Hc*KCRs[14,15] and ChRmine, the only cryptophyte BCCR with available structure[16,17]. Residues conserved in *Hc*KCR1, *Hc*KCR2 and *Hc*CCR (Fig. 2c, Supplementary Fig. 1) stabilize the trimers by polar interactions between two protomers via the side chains of Asp48 in TM1 with Arg128 (TM4) near the cytoplasmic surface, Thr79 (TM2) with Trp161 (TM5) in the middle of the membrane, and Asp90 (TM2) with Ser148 (TM4–TM5 loop) on the extracellular surface. In addition, the side chain of Tyr154 (TM5) forms H-bonds with main chain carbonyls of Ala83 and Phe96, and the side chain of Glu65 (TM2) with main chain nitrogens of Lys126 and Ile127 in TM4 (Leu127 in *Hc*CCR and *Hc*KCR2). Among these residues, only Glu65 is highly conserved in *Hc*KCR homologs and BCCRs. In ChRmine, the Asp65 homolog (Asp69) is also involved in trimer formation, but its interaction partner is the side chain of Ser138[16], which is conserved in most BCCRs but replaced with Arg128 in *Hc*ChRs. Interestingly, the side

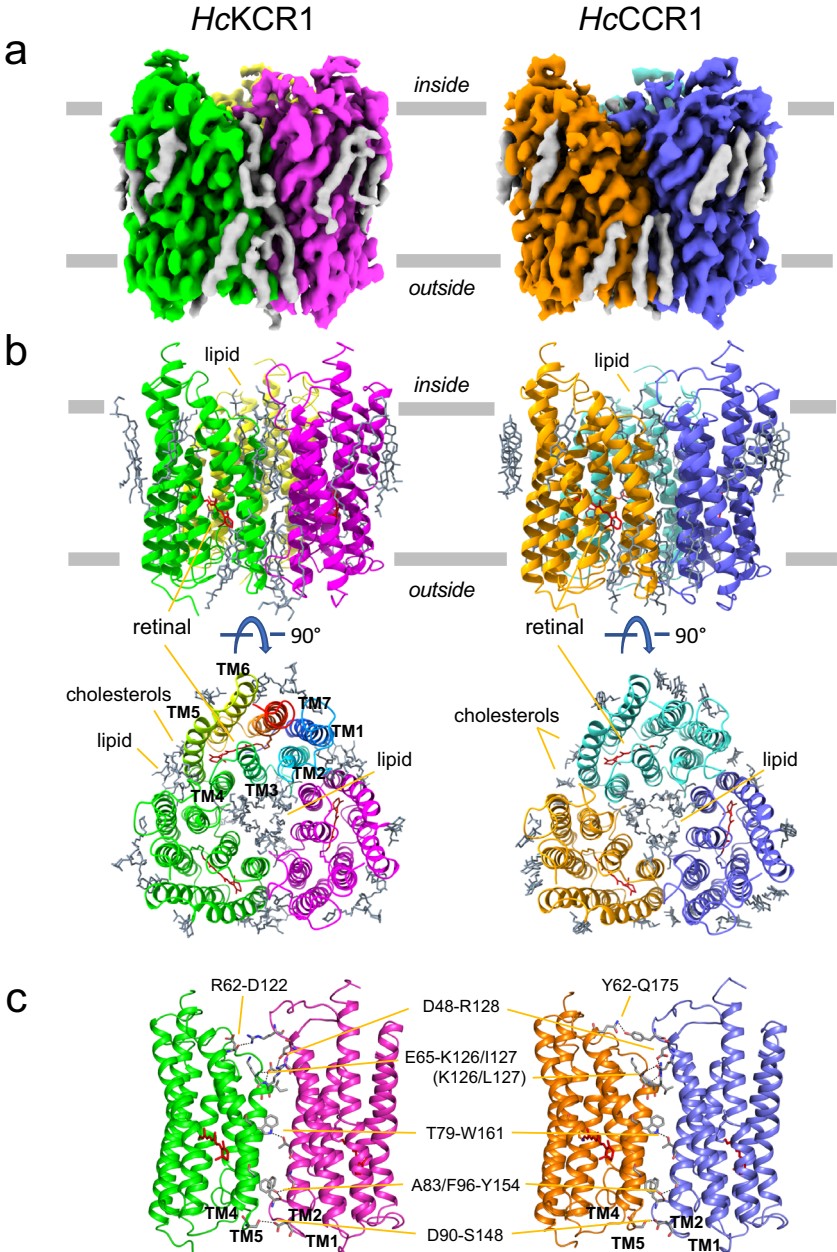

**Fig. 2 | Cryo-EM structures of *Hc*KCR1 and *Hc*CCR. a** Cryo-EM density maps of the *Hc*KCR1 and *Hc*CCR homotrimers viewed from the membrane plane. Protomers are colored in magenta, green, and yellow for *Hc*KCR1, and in blue, orange, and turquoise for *Hc*CCR, respectively. **b** Structure models viewed from the membrane plane (upper) and intracellular side (lower) with all-*trans*-retinal shown in red. In one protomer of *Hc*KCR1 trimer helices are shown in different colors and labelled to show the order of transmembrane helices. **c** Interprotomer connecting residues observed in the trimeric configuration.

chain of Arg62 in the TM1–TM2 loop of *Hc*KCR1 also interacts with the main chain carbonyl of Asp122 (TM3) on the neighboring protomer, whereas in *Hc*CCR side chains of Tyr62 and Gln175 (TM5) interact (Fig. 2c). The switching of binding partner might be responsible for the different positioning of functionally important residues in TM3 in the putative ion conducting pathway, such as Asp116 (see below).

The central pore between the three *Hc*KCR1 protomers is filled with six phospholipids, which we modeled as dioleoylphosphatidylethanolamine (DOPE), three in each leaflet of the bilayer, that prevent ions from translocating through the pore. In addition, the membrane-facing outer surface of the *Hc*KCR1 trimer is decorated with nine sterol lipids and nine phospholipids, which we modeled as DOPE molecules, although in some cases the density did not clearly exclude

choline headgroups and different hydrocarbon chains. As no phospholipids were added during protein purification, DOPE must have been carried over from *Pichia* membranes. Sterol lipids were modeled as cholesterol and may originate from the host membranes. Cholesteryl hemisuccinate (CHS) was supplemented during purification, raising the possibility that some sterol lipids are CHS molecules, but density for hemisuccinate is weak or missing. In the case of the *Hc*CCR trimer model, the central plug of six DOPE lipids is also observed. On the outer surface of the *Hc*CCR trimer 21 sterol lipids are bound, which all were modeled as cholesterols.

Previously, we have used ColabFold software to create homology models of *Hc*KCR1 and *Hc*CCR[5]. RMSD of Cα atoms between the model and the structure was 0.60 Å for *Hc*KCR1 and 0.61 Å for *Hc*CCR. The

**Table 1 | Cryo-EM data collection, refinement, and validation statistics**

| | *Hc*KCR1 | *Hc*CCR |
|---|---|---|
| Data collection and processing | | |
| Microscope | FEI Titan Krios | FEI Titan Krios |
| Detector | Falcon 4i | Falcon 4i |
| Magnification (Å) | 75,000 | 75,000 |
| Voltage (kV) | 300 | 300 |
| Electron exposure (e⁻/Å²) | 40 | 40 |
| Defocus range (µm) | 0.8–2.4 | 0.8–2.4 |
| Pixel size (Å) | 1.03 | 1.03 |
| Symmetry imposed | C3 | C3 |
| Initial particle images (no.) | 2,685,194 | 2,674,318 |
| Final particle images (no.) | 297,015 | 296,840 |
| Map resolution (Å) | 2.88 | 2.84 |
| FSC threshold | 0.143 | 0.143 |
| Map resolution range (Å) | 2.4–3.3 | 2.4–3.2 |
| Refinement | | |
| Initial model used | ab initio | ab initio |
| Map sharpening B factor (Å²) | −158.5 | −153.1 |
| Model composition (monomer) | | |
| Nonhydrogen atoms | 2330 | 2284 |
| Protein residues | 240 | 240 |
| Lipids | 8 | 9 |
| Water | 15 | 15 |
| B factors (Å²) | | |
| Protein | 65.80 | 58.81 |
| Lipids | 71.57 | 62.23 |
| Water | 68.73 | 54.52 |
| R.m.s. deviations | | |
| Bond lengths (Å) | 0.003 | 0.003 |
| Bond angles (°) | 0.449 | 0.447 |
| Validation | | |
| MolProbity score | 1.06 | 1.20 |
| Clashscore | 2.50 | 1.73 |
| Poor rotamers (%) | 0 | 0 |
| Ramachandran plot | | |
| Favored (%) | 97.90 | 95.38 |
| Allowed (%) | 2.10 | 4.62 |
| Disallowed (%) | 0 | 0 |

models were particularly good for helices TM1, TM6, and TM7 (Supplementary Fig. 7). However, the homology models were weak in the extracellular TM2-TM3 loop region and did not predict partial unwinding of TM3 at the extracellular end, an unusual feature of *Hc*ChRs that they share with ChRmine[16,17] (Supplementary Fig. 7). While we focus here on *Hc*KCR1 and *Hc*CCR, we provide in the supplement a comparison of *Hc*KCR1 with ChR2 and ChRmine (Supplementary Figs. 8–10, Supplementary Discussion).

**Internal cavities in a protomer indicate the putative cation conduction pathway**

The overall closed-state (dark) structures of *Hc*KCR1 and *Hc*CCR protomers are almost identical, but local variations give insight into the functional differences. In both structures a series of water-containing cavities separated by three constrictions is found between TM1, TM2, TM3, and TM7 (Fig. 3a, b). These cavities, the shapes, volumes and electrostatic potentials of which differ in the two proteins, presumably merge upon illumination to form a continuous cation conduction pathway. Small spherical densities within the cavities were interpreted as water molecules except those that show electrostatic interactions with aromatic systems and lack hydrogen bonding (H-bonding), which were interpreted as Na⁺ buffer component (Supplementary Fig. 11). The *Hc*KCR1 monomer model contains 15 water molecules and one Na⁺, whereas the *Hc*CCR monomer contains 15 water molecules and three Na⁺. All densities interpreted as Na⁺ ions are found outside of the putative cation conduction pathway (Fig. 3a, b) and therefore are unlikely to be due to transported cations trapped in the pore of the closed channel.

The side chain H-bond graphs computed for *Hc*KCR1 and *Hc*CCR using the Bridge[21] and C-Graphs[22] software provide an overview of the location of the internal H-bonding network of the protein, and directly illustrate the relationship between amino acid residue sequence and the local H-bonding network. With a 4 Å distance criterion the H-bond graph of *Hc*KCR1 contains in total 66 H-bonds, of which 24 are water-side chain interactions (Fig. 3c; Supplementary Fig. 12); the graph of *Hc*CCR has overall fewer H-bonds, 50, and 17 from these are formed between a water molecule and an amino acid residue (Fig. 3d; Supplementary Fig. 13).

The putative cation conduction pathway can be divided into the intracellular segment, the photoactive site and the extracellular segment containing a cluster of aromatic residues (Fig. 3a, b), each with characteristic H-bonding patterns. Below we explore this pathway in detail by following the putative efflux path of a K⁺ ion.

**Intracellular segment of the cation conduction pathway**

Three of the five residues identified by functional studies[5] as critical determinants of K⁺ or Na⁺ selectivity (Leu/Phe69, Ile/Ser73, and Asp116) are located in the intracellular segment of the putative cation conduction pathway. It is constricted by a helix-linking H-bonding network including water molecules and residues Ser70 in TM2, Asp116 and Thr120 in TM3, and Arg244 in TM7, all of which are conserved in both *Hc*ChRs (Fig. 4a, b); in addition, Asn67 H-bonds to Glu248, and Glu246 to Tyr53 (Fig. 3c, d). Some residues that are conserved in *Hc*KCR1 and *Hc*CCR nevertheless have distinct H-bond environments. Examples include Thr120, which has shorter H-bonds to both Asp116 and Arg244 in *Hc*CCR than in *Hc*KCR1, and Glu248, which bridges via water to Tyr58 in *Hc*KCR1 but not in *Hc*CCR (Fig. 3c, d).

Asp116 corresponds to Asp96 in BR, the proton donor to the RSB during the BR proton pumping cycle. In both *Hc*KCR1 and *Hc*CCR, the side chains of Asp116 and Arg244 are interacting. The distance between these side chains is 3.4 Å for *Hc*KCR1 and 3.0 Å for *Hc*CCR, respectively (Supplementary Figs. 12 and 13). The density of Asp116 is not well-defined arguing for flexibility and ease of breaking its interaction with Arg244 to open up the constriction for cation passage. Although Asp116 is conserved in *Hc*KCR1 and *Hc*CCR, its mutagenetic neutralization in *Hc*KCR1 results in conversion of the K⁺-selective channel into a Na⁺-selective one with a much smaller conductance[5,6,14,15]. Comparison of the closed-state (dark) *Hc*KCR1 and *Hc*CCR structures suggests an explanation for this conversion. In *Hc*KCR1, the Asp116 side chain is H-bonded to two water molecules close to Thr120 and Ile73 in the nearby cavities (Fig. 4a). In *Hc*CCR, the water molecule is H-bonded to Ser73 instead, and the other water is missing. This is likely caused by the bulky aromatic side chain of Phe69 located in the interface between TM2 and TM3 (Leu69 in *Hc*KCR1), which also influences the position of Asp116 (Fig. 4b). In *Hc*CCR, Ser73 and Trp199 each H-bond to a water molecule (Fig. 3d); in *Hc*KCR1, Ile73 cannot bind a water and the H-bond graph reveals instead two unique inter-helical interactions between Ser39 and Ser234, and between Met43 and Cys74 (Fig. 3c).

The S73I mutation increased K⁺ selectivity of *Hc*CCR[5], indicating that the polar side chain protruding into the cavity is a determinant for Na⁺ selection. This conclusion is confirmed by an increase in Na⁺ selectivity in the *Hc*KCR1_I73S mutant revealed by a shift of the reversal

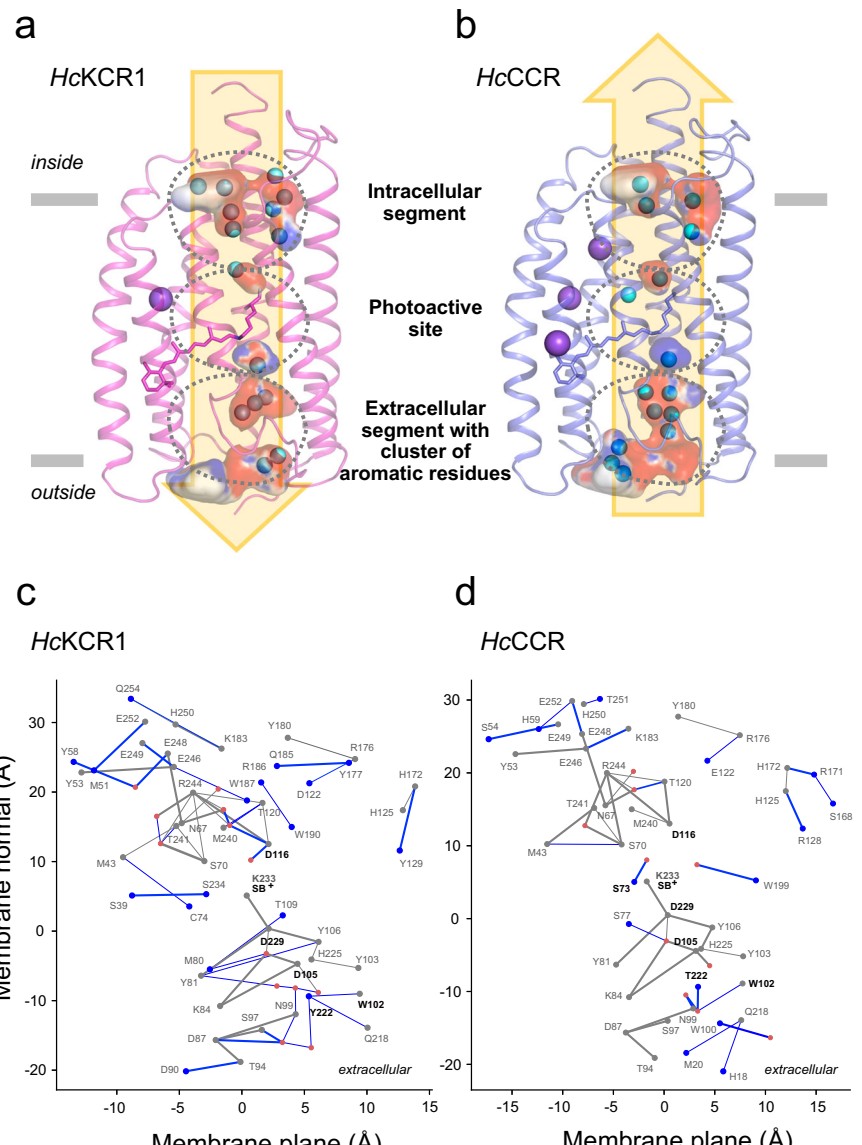

**Fig. 3 | Comparison of cavities, cation conduction pathway, and H-bonding networks.** Comparison of the internal cavities of *Hc*KCR1 (**a**) and *Hc*CCR1 (**b**) and their electrostatic potential. Cavities were modelled with HOLLOW[50] and electrostatic map colored with APBS[49] software. All-*trans*-retinal and Lys233 are shown as stick model. **c, d** Difference H-bond graphs illustrate the conserved and unique H-bonds of *Hc*KCR1 and *Hc*CCR. Gray dots (graph nodes) and lines (graph edges) indicate amino acid residues and their connections which are conserved (present) in both structures, whereas blue nodes and edges show amino acid residues and H-bonds which are unique to each structure. Red dots (nodes) are water molecules determined in the cryo-EM structures. The H-bond graphs were computed using Bridge[21] and C-Graphs[22] programs with a distance criterion of 4 Å between the H-bond donor and acceptor hetero-atoms. Bold edges indicate distances ≤ 3.5 Å. The H-bond graph is projected to a two-dimensional plane, where the vertical axis corresponds to the coordinates of the Cα atoms of the amino acid residues along the membrane normal (z coordinate), and the horizontal axis shows the Principal Component Analysis (PCA) projection along the membrane plane (xy coordinates). **c** Difference H-bond graph of *Hc*KCR1 relative to the conserved H-bond graph. **d** Difference H-bond graph of *Hc*CCR relative to the conserved H-bond graph.

potential ($V_{rev}$) to more depolarized values compared to the wild-type (WT) *Hc*KCR1 (Fig. 4c). A possible mechanism for this selectivity is that in *Hc*CCR, Ser73 directs its side chain towards the carboxylate group of Asp116. The distance between the OH group of Ser73 and the COOH group of the Asp116 side chain is 6.2 Å. It is possible that in the open state, these two residues stabilize a Na$^+$ ion in between them, whereas this tight coordination may be unfavorable for K$^+$, thereby substitution of Ile73 with a serine diminished K$^+$ selectivity. Consistent with this hypothesis, Thr in this position produced no effect compared to Ile in either channel (Fig. 4c), indicating that the hydroxyl group must be properly located within the cavity to confer Na$^+$ selectivity. The I73D and I73F mutations in *Hc*KCR1 caused depolarizing $V_{rev}$ shifts compared to the WT, but the corresponding mutations in *Hc*CCR

completely abolished channel currents (Supplementary Fig. 14), so the effects of these substitutions on selectivity could not be measured.

In the *Hc*KCR1 structure, the presence of Ile73 and Leu69 on the helical interface between TM2 and 3 moves TM2 away by 1.4 Å (measured at the Cα atom of Val66). This helical movement slightly enlarges the vestibule, which may help accommodate a large K$^+$ ion in the vicinity. It is worth noting that these subtle features also induce noticeable alteration in the electrostatic profile in the tunnel as evidenced by the altered water coordination at the D116 position. We anticipate these changes collectively contribute to the distinct substrate selectivity between *Hc*KCR1 and *Hc*CCR.

At the cytoplasmic surface, we note a local cluster of four Glu (Glu246, Glu48, Glu49, and Glu252) and one (His250, *Hc*KCR1) or two

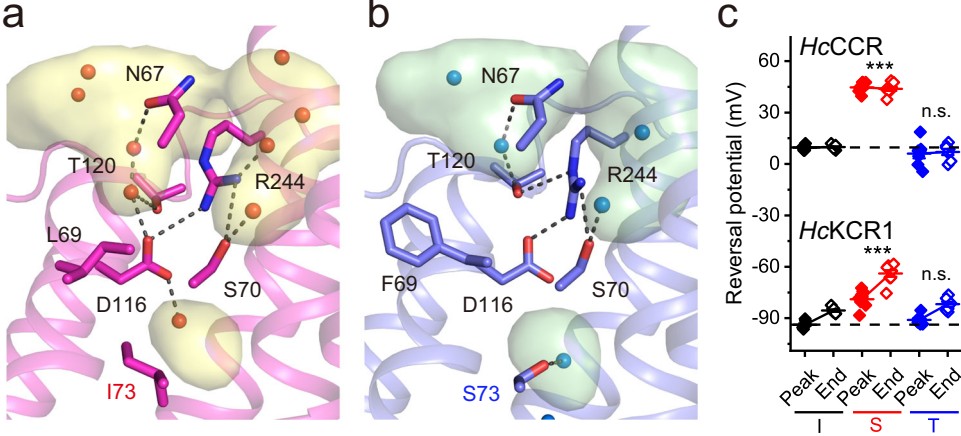

**Fig. 4 | Intracellular segment.** Intracellular segment of *Hc*KCR1 (**a**) and *Hc*CCR (**b**) colored in magenta and blue, respectively, with the cavities colored in yellow and green. Key residues are displayed as stick models. Water molecules are represented by spheres. The black dashed lines indicate H-bonds. The internal cavities modelled with HOLLOW[50] are shown. The backbone of TM 2 is omitted for clarity. **c** The reversal potentials ($V_{rev}$) of channel currents measured in the variants with Ile, Ser or Thr at the residue position 73. Wild-type *Hc*KCR1 contains Ile, and wild-type *Hc*CCR, Ser at this position. The symbols (black for the I mutants, red for the S mutants and blue for the T mutants; filled for the peak currents and empty for the end currents) are the mean values, the error bars, s.e.m. ($n = 8$ cells for *Hc*KCR1_I73S and WT *Hc*CCR, and 7 cells for other variants). $p = 2.97E\text{-}07$ (peak currents) and $5.20E\text{-}09$ (end currents) by one-way ANOVA with the Tukey correction for multiple comparisons (alpha 0.0001 for *Hc*KCR1 currents and 0.05 for *Hc*CCR currents) to the respective I variants. n.s., not significant. Source data are provided as a Source Data file.

His (His59 and His250, *Hc*CCR). Such bulk-exposed clusters of Glu and His sidechains are hypothesized to function as proton-collecting antennas[23–25] and may be important for the proton conductance of these channels and/or act as antennas for Na$^+$ and K$^+$.

**Photoactive site**

As in all microbial rhodopsins[26], the retinal chromophore in *Hc*KCR1 and *Hc*CCR is bound via a protonated retinylidene Schiff base (RSB$^+$) linkage to a conserved lysine residue in TM7 (Lys233). For both proteins, the chromophore density could be modeled with all-*trans*-retinal and the β-ionone ring in the coplanar (6-s-*trans*) configuration with respect to the polyene chain. This is consistent with the results of Fourier-transformed (FT) Raman spectroscopy of purified *Hc*CCR and *Hc*KCR1 (Supplementary Fig. 15). The fingerprint C-C stretching region of the FT-Raman spectra reports on the retinal configuration. As follows from the strong peaks at ~1165 and 1203 cm$^{-1}$, the retinal configuration is predominantly all-*trans* in both proteins. The presence of the 1180 cm$^{-1}$ band (labeled orange) suggests that a smaller fraction of the protein may bind 13-*cis*-retinal (consistent with the earlier HPLC results for *Hc*KCR1[14]). The 13-*cis*-retinal fraction is likely higher in *Hc*KCR1 than in *Hc*CCR, as suggested by the larger relative amplitude of the 1180 cm$^{-1}$ band. As the 13-*cis*-retinal-bound forms of *Hc*ChRs are likely non-electrogenic, this may explain a larger blue shift of the absorption maximum of detergent-purified *Hc*KCR1 from the spectral maximum of its photocurrents (to 522 nm from 540 nm), as compared to that in *Hc*CCR (to 521 nm from 530 nm). The Raman spectra of *Hc*KCR1 and *Hc*CCR show that the polyene chain of the chromophore is not twisted, as no strong hydrogen-out-of-plane vibrations are observed in the 900–1000 cm$^{-1}$ region (Supplementary Fig. 15).

The RSB$^+$ separates the internal cavities located inwardly and outwardly of it and thus represents the key structural element constricting the cation conduction pathway in the dark state of *Hc*KCR1 and *Hc*CCR (Fig. 5a, b). The constriction involves a salt bridge between the RSB$^+$ and its counterion Asp229 in TM7, which is in the center of a H-bonding network with Tyr81 in TM2 and Tyr106 in TM3. This arrangement is also found in the BCCR ChRmine, in which the corresponding residues Asp253, Tyr85 and Tyr116 are employed[16]. In contrast, the Schiff base of BR is connected via a water molecule to the H-bonding network (Fig. 5c). Light-induced all-*trans* to 13-*cis* retinal

isomerization is expected to break this interhelical H-bonding network to enable formation of the cation conduction pathway. In proton-pumping BR, Asp212 and Tyr57 correspond, respectively, to Asp229 and Tyr81 of *Hc*ChRs, and the locations and H-bonding of these residues are very similar in all three proteins (Fig. 5a–c). However, in BR, a Tyr in TM6 (Tyr185) rather than in TM3 (Tyr106 in *Hc*ChRs) H-bonds to the RSB$^+$ counterion Asp212 in TM7 (Fig. 5c). Tyr106 and His225 are within 3.6 Å distance in *Hc*KCR1, as compared to 3.4 Å in *Hc*CCR (Supplementary Figs. 12 and 13). In both *Hc*KCR1 and *Hc*CCR at pH 7.4, the Y106A mutations produced a larger spectral shift than the Y81A and Y106F mutations (Fig. 5d, e). In the D229N mutants of both *Hc*ChRs, channel currents could only be detected at the holding voltages far away from $V_{rev}$, so their $V_{rev}$ values could not be determined accurately. The Y81A and Y106A mutations produced statistically significant $V_{rev}$ shifts in *Hc*KCR1 (and no shifts in *Hc*CCR), but these shifts were very small (Fig. 5f), indicating that the interhelical Tyr81-Asp229-Tyr106 H-bonding network does not play a major role in determination of channel selectivity. However, this network is critical for channel gating, as the Y81A and Y106A mutations each strongly inhibited channel currents in both *Hc*ChRs (Supplementary Fig. 16). The Y106F mutations produced less photocurrent inhibition, as compared to the Y106A mutations (Supplementary Fig. 16), which suggests that π-π interaction of the aromatic side chain in this position with the retinal chromophore is required for channel function.

Asp105 is the second carboxylate at the photoactive site corresponding to Asp85 of BR, located in both *Hc*ChRs at a larger distance from the RSB$^+$ than Asp229 (Fig. 5a, b). Within a water molecule distance from Asp229, Asp105 salt-bridges to Lys84 and connects to the water-mediated H-bonding network of Asn99 at the extracellular side of the protein (Fig. 3c, d). The Asp105-Asp229 H-bond and some of the other H-bonds of Asp105 and Asp229 are present in the networks of both proteins, but the H-bond environment of Asp105 has additional features specific to either structure. In *Hc*KCR1, Asp105 is close to Met80 and Thr109—this latter residue being within one helical turn of Tyr106 (Fig. 3c); Asp105 of *Hc*CCR is instead close to Ser77, and the H-bond distance between Tyr106 and His225 is shorter (Fig. 3d). Mutagenetic neutralization of Asp105 completely abolished channel currents in both *Hc*KCR1 and *Hc*CCR (Supplementary Fig. 16), suggesting the importance of this residue for channel function.

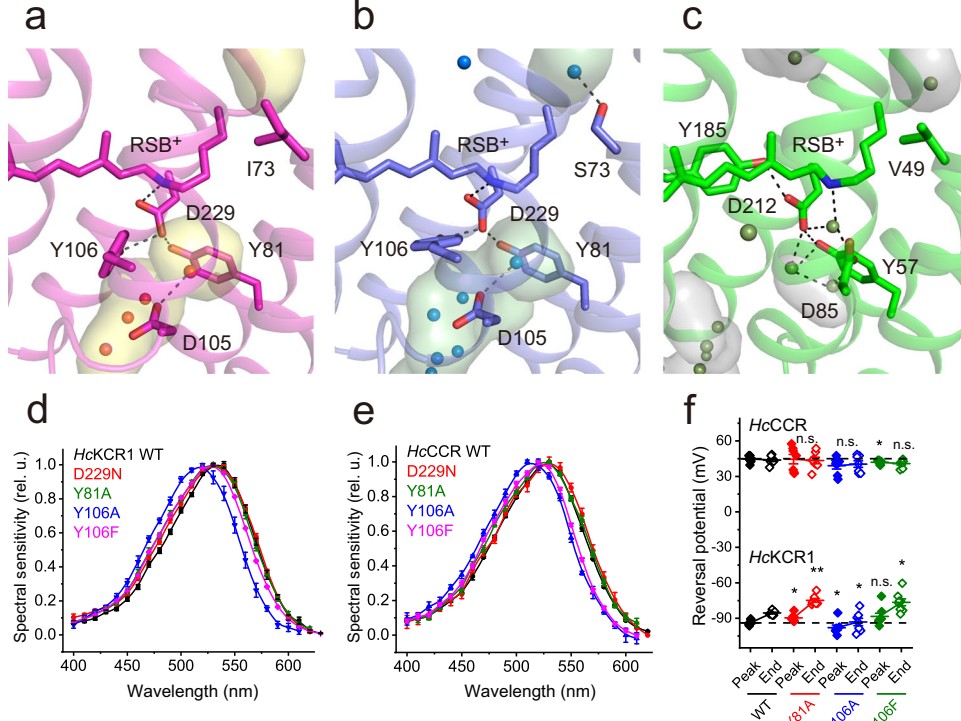

**Fig. 5 | Photoactive site.** Photoactive site near the retinal Schiff base (RSB⁺) of *Hc*KCR1 (**a**), *Hc*CCR (**b**), and Bacteriorhodopsin (**c** BR, PDB ID: 1C3W) colored in magenta, blue, and green, respectively, with the cavities colored in yellow, green, and grey. Key residues lining the cavities are displayed as stick models. Water molecules are represented by spheres. The black dashed lines indicate hydrogen bonds. The internal cavities modelled with the program HOLLOW[50] are shown. The action spectra of photocurrents by *Hc*KCR1 (**d**) and *Hc*CCR (**e**) variants. The symbols are the mean values, the error bars, s.e.m. (*n* = 16 cells for *Hc*CCR_D229N, 8 cells for *Hc*CCR_Y106F, and 6 cells for other variants). **f** The reversal potentials (V_rev)

of channel currents. The symbols (black for the wild types, red for the Y81A mutants, blue for the Y106A mutants and green for the Y106F mutants; filled for the peak currents and empty for the end currents) are the mean values, the error bars, s.e.m (*n* = 8 cells for *Hc*KCR1_Y106F and WT *Hc*CCR, and 7 cells for other variants). *p* = 0.02, 0.03 and 0.06 (peak currents) and 0.002, 0.04, and 0.01 (end currents) for *Hc*KCR1_Y81A, Y106A and Y106A, respectively; 0.82, 0.08 and 0.04 (peak currents) and 1, 0.45, and 0.1 (end currents) for *Hc*CCR_Y81A, Y106A, and Y106A, respectively by the two-sided Mann-Whitney test compared to the respective WT. n.s., not significant. Source data are provided as a Source Data file.

The retinal binding pocket defines light absorption properties of rhodopsins together with the electrostatic potential around the RSB⁺[26,27]. Supplementary Fig. 17 shows that the pockets of *Hc*KCR1 and *Hc*CCR are nearly identical, which is consistent with their similar spectral sensitivities (the maxima of the photocurrent action spectra are 540 and 530 nm, respectively[4,5]). Thr136 and Gly140 in TM4 and Pro203 and Phe206 in TM6 embed the β-ionone ring. In *Hc*KCR2, the second KCR from *H. catenoides*, Thr136 and Gly140 are replaced with Ala residues, which twists the β-ionone ring with respect to the polyene chain[14] and explains the observed large blue shift of the *Hc*KCR2 spectral maximum (490 nm) relative to *Hc*KCR1 and *Hc*CCR[4].

**Extracellular segment with a cluster of aromatic residues**
Moving further outward along the putative cation conduction pathway from the RSB towards the extracellular opening, a cluster of aromatic residues is found that acts as another constriction in *Hc*KCR1 and *Hc*CCR (Fig. 6a, b). In BR, the corresponding residues form the proton release network contributed by H-bonded water molecules and charged and polar residues, including Arg82, Ser193, Glu194 and Glu204 (Fig. 6c)[28,29]. In particular, Arg82 of BR is replaced with Trp102; Ser193 with Trp210; and Glu204 with Phe221 in both *Hc*KCR1 and *Hc*CCR. Thr205 in BR is analogous to Thr222 in *Hc*CCR and replaced with Tyr222 in *Hc*KCR1. In *Hc*KCR1 Tyr222 forms H-bonds to Trp102 and Gln218. As a result, the extracellular channel opening is interrupted with the bulky aromatic Tyr side chain in *Hc*KCR1 but continues deeper into the molecule in *Hc*CCR, creating a key determinant for K⁺ selectivity. Analysis of the closed-state (dark) *Hc*ChR structures with the program CAVER confirmed an extracellular tunnel for *Hc*CCR

where cavities are merged, similar to C1C2 ChR[30], but lack of an extracellular tunnel for *Hc*KCR1 where cavities are separated and the H-bonding network of Tyr222 is expected to be altered by retinal isomerization (Supplementary Fig. 18). Similarly, alteration of a H-bonding network was postulated for fast channel closing in *Guillardia theta* anion channelrhodopsin 1 (*Gt*ACR1)[31]. Prior mutagenesis studies have revealed that Trp102 and Tyr222 are required for K⁺ selectivity in *Hc*KCR1, as their replacement with non-aromatic residues decreased P_K/P_Na[5,6,14]. Moreover, the presence of both Trp102 and Tyr222 is required for K⁺ selectivity of natural KCR homologs, as channels in which even one of these residues is not conserved are not K⁺ selective[4,5]. In *Hc*KCR1, the Y222W mutation caused a stronger V_rev shift to more depolarized values indicating a decrease in P_K/P_Na than that caused by the W102Y mutation (Fig. 6d and Supplementary Fig. 19), which suggests that the residue in the position 222 is more important for K⁺ selectivity than that in the position 102. Swapping of the two residues (the W102Y_Y222W double mutant) did not restore the WT phenotype but showed the same V_rev as the Y222W mutation alone, which confirmed this conclusion. Tyr222 in *Hc*KCR1 is in the center of a H-bonding network with Gln218 and Trp102, and, via two water molecules, to Asp87, which further connects to Asn99 (Fig. 6a). Replacement of Tyr222 with Thr in *Hc*CCR disrupts this network, and Thr222 bridges via two water molecules to Asn99, and Trp102 is within 4 Å of this water bridge (Fig. 6b). A possible need for the precise location of Tyr222 is its potential dehydration of K⁺ by π-electron cloud interaction. Functional importance of this structural difference is confirmed by the observations that mutations of Asp87, Asn99, or Gln218 decrease K⁺ selectivity of *Hc*KCR1[6,14,15].

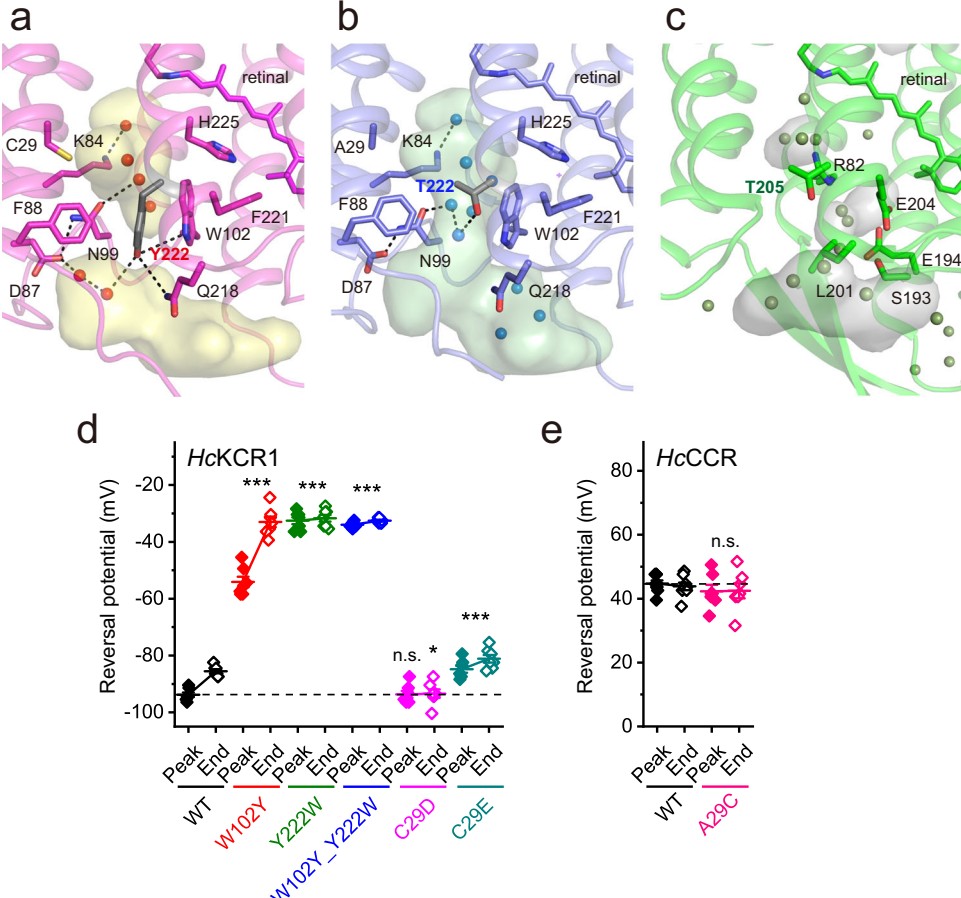

**Fig. 6 | Extracellular aromatic cluster.** Extracellular regions of *Hc*KCR1 (**a**), *Hc*CCR (**b**), and bacteriorhodopsin (**c**, PDB ID: 1C3W) are colored in magenta, blue, and green, respectively, with the cavities colored in yellow, green, and grey. Key residues lining the cavities are displayed as stick models. Presence of Y222 instead of T222 in *Hc*KCR1 occludes the channel pore compared to *Hc*CCR. Cavities were calculated with the program HOLLOW[50]. The reversal potentials ($V_{rev}$) of channel currents measured in the mutants of extracellular residues in *Hc*KCR1 (**d**) and

*Hc*CCR (**e**). The symbols (colors indicate individual mutants; filled for the peak currents and empty for the end currents) are the mean values, the error bars, s.e.m ($n$ = 8 cells for WT *Hc*CCR, and 7 cells for other variants). $p$ = 3.21E-33 (*Hc*KCR1 peak currents), 3.12E-33 (*Hc*KCR1 end currents), 2.82E-01 (*Hc*CCR peak currents) and 0.59 (*Hc*CCR end currents) by one-way ANOVA with the Tukey means comparison to the respective WTs. n.s., not significant. Source data are provided as a Source Data file.

Cys29 in TM1 of *Hc*KCR1 is located just inward of the aromatic cluster (Fig. 6a) and is replaced with Asp47 in *Wi*ChR, a KCR homolog from *Wobblia lunata*. *Wi*ChR shows higher $P_K/P_{Na}$ than *Hc*KCR1, which is reduced by the D47C mutation[6]. Under physiological (asymmetric) ionic conditions $V_{rev}$ shifts towards more depolarized values during continuous illumination in all tested KCRs[4–6]. $V_{rev}$ values measured at the peak time were equal in the *Hc*KCR1_C29D mutant and the WT, but the mutant did not exhibit the $V_{rev}$ shift during illumination (Fig. 6d). The C29E mutation decreased K+ selectivity both at the peak time and at the end of illumination. Cys29 is replaced with Ala in *Hc*CCR. Its mutation to Cys did not change K+ selectivity of this channel (Fig. 6e), which is consistent with our previous results obtained upon replacement of the entire TM1 of *Hc*CCR with that of *Hc*KCR1[4].

## Discussion

Our structures of peptidisc-embedded *Hc*KCR1 and *Hc*CCR are those of the closed channels, but nevertheless they suggest an explanation for the dramatic difference in the cation selectivity (i.e., relative permeability) between these channels under illumination revealed by patch clamp experiments[4,5,32]. At least two regions of the putative cation conduction pathway determine K+ selectivity of *Hc*KCR1 and Na+ selectivity of *Hc*CCR. The first region is located just inward of the cytoplasmic entry to the channel pore and is centered on conserved Asp116, the homolog of the proton donor Asp96 in BR. Molecular

dynamics (MD) simulations in *Hc*KCR1 revealed transient binding of K+ to Asp116 accompanied by the loss of the salt bridge between Asp116 and Arg244, suggesting that Asp116 is involved in partial dehydration of K+ entering from the cytoplasmic side[14]. Comparison of *Hc*KCR1 and *Hc*CCR structures shows different orientations and H-bonding patterns of Asp116 caused by the replacement of Leu69 of *Hc*KCR1 with Phe in *Hc*CCR, and Ile73, with Ser (Fig. 4), suggesting that the Asp116-Arg244 salt bridge disruption is involved in the cation dehydration. In *Guillardia theta* CCR2 (*Gt*CCR2), a cryptophyte BCCR studied in detail, deprotonation of the Asp116 homolog (Asp98) is required for cation channel opening and occurs >10-fold faster than reprotonation of the RSB, which kinetically correlates with channel closing[10].

The second region of the conduction pathway critical for K+ selectivity is the extracellular aromatic cluster (Fig. 6). Mutations of Phe88, Trp102, Phe221, and His225 reduce K+ selectivity of *Hc*KCR1[5,6,14,15], but the most important difference between K+ selective *Hc*KCR1 and Na+ selective *Hc*CCR is the presence of Tyr or Thr, respectively, in the residue position 222. Replacement of Tyr with Thr leads to extension of the water-filled extracellular vestibule towards the Schiff base and rearranges the H-bonding network in the region (Fig. 6). When the channel opens under illumination, Tyr222 (likely assisted by Phe221 and His225 on the other side of the putative conduction cation pathway) allows K+ passage but functions as a barrier to the flux of larger hydrated cations like Na+ and divalent cations.

In contrast, the presence of Thr at position 222 widens the channel to enable passage of larger hydrated Na$^+$. Similarly, mutation of Trp102 in the center of the aromatic cluster to a non-aromatic residue disrupts the electrostatic interactions within this cluster, making it porous for Na$^+$ and thereby destroys this K$^+$ selectivity filter. Interestingly, also the CNGA1/CNGB1 cyclic nucleotide-gated channel, which carries Na$^+$/Ca$^{2+}$ inward currents, employs similar elements−an Arg residue and an aromatic cluster−for gating. An arginine residue of the single CNGB1 subunit reaches in the intracellular segment into the ionic pathway to block the pore, thus introducing an additional gate. This gate is different from the central hydrophobic gate which is made up of four aromatic residues[33].

The residues at the photoactive site (Tyr81, Asp105, Tyr106, and Asp229) appear to be essential for channel gating in both HcKCR1 and HcCCR, as their neutral substitutions strongly inhibit channel currents (Fig. 6 and Supplementary Fig. 16). Asp229 in both HcChRs, and the homologous Asp212 in BR, are H-bonded to two Tyr residues. While one of them is in TM2 (Tyr81 in HcChRs and corresponding Tyr57 in BR), the second one is contributed by TM3 (Tyr106 in HcChRs) or TM6 (Tyr185 in BR). Such an H-bonding pattern involving a Tyr in TM3 as in HcChRs is also observed in the inward proton pump schizorhodopsin 4 (SzR4)[34], where Tyr71 corresponds to Tyr106 of HcChRs, although overall sequence homology between KCRs and SzRs is very low. The direct bonding of Asp229 to RSB in HcChRs is in contrast to the outward-directed proton pumps, in which the homologous groups are connected via a strongly hydrogen-bonded water molecule.

A shift of $V_{rev}$ to more depolarized values observed in KCRs during illumination is little understood and decreases their utility as optogenetic silencers. One possible reason could be transient elevation of extracellular K$^+$ concentration as detected in neurons expressing C. reinhardtii ChR2_H134R[35]. However, the disappearance of this $V_{rev}$ shift after replacement of extracellular Na$^+$ with Ca$^{2+}$ or Mg$^{2+}$ (ref. [4]) suggests that it is caused by accumulation of a photocycle intermediate with a higher selectivity for Na$^+$ than that of the primary conductive state. Elimination of the $V_{rev}$ shift by the C29D mutation (Fig. 6d) and its increase by the W102Y mutation (Fig. 6d) shows that the mutated residues regulate formation of the Na$^+$ selective state.

The canonical K$^+$ selective channels function with a tetrameric selectivity filter that fully dehydrates K$^+$(aq) ions, which then can be translocated through the narrow pore. The backbone carbonyls of the filter are spaced in a way to match the positions of the eight waters in a hydrated K$^+$(aq) ion. Na ions in contrast have a lower hydration number of 6, and the water molecules in the hydration shell are not energetically favorable to be stripped from the ion. While the extracellular cavities in the closed structure of HcKCR1 are smaller than in HcCCR, it is unknown if HcKCR1 is capable of dehydrating K$^+$(aq) ions as in the prokaryotic tetrameric channel KcsA. HcKCR1 does not have any inward facing backbone carbonyls that are not a part of an alpha helix. Our study provides structural insights into the dramatic difference in ion selectivity between HcKCR1 and HcCCR observed despite their high degree of residue conservation, which will be helpful for further development of optogenetic tools for inhibiting neuronal spiking. Detailed characterization of the photochemical reaction cycle and structural analysis of the channel open state can be envisioned as the next steps toward this goal.

In addition to the structural insights, this study illustrates the efficiency of using peptidiscs. In 2018, peptidiscs were introduced as a novel method for stabilizing solubilized membrane proteins[18]. Their application in cryo-EM structure analysis has so far been limited to few studies[36–39], which is in contrast to the frequent use of nanodiscs. In our study we made use of several advantages of peptidiscs. These include: i) the ease of gentle and quick exchange of detergent for peptidisc peptide, which limits the time detergent-sensitive membrane proteins are outside a membrane environment[40], ii) the obtained monodispersity of the membrane protein/peptidisc particles

(Supplementary Fig. 4), and iii) the lack of peptidiscs without embedded protein, which simplifies purification. In the present study we were able to determine structures of HcChR 7TM domains, which showed high resolution on par with structures of ChRs embedded in nanodiscs[14–17]. We foresee a wider applicability of peptidiscs in high resolution cryo-EM studies of rhodopsins and GPCRs and membrane proteins beyond the 7TM scaffold. Given the described advantages, peptidiscs appear to be suited to accelerate the throughput of structure determination in basic as well as applied science, such as drug discovery.

## Methods

### Molecular biology and bioinformatics
The polynucleotides encoding the amino acid residues 1–265 of HcKCR1 and HcCCR (Genbank accession numbers MZ826861 and OL692497, respectively) were fused with the C-terminal 8His-tag and cloned in the pPICZalpha-A vector (Invitrogen) for expression in Pichia pastoris, or fused with the C-terminal mCherry tag and cloned in the pcDNA3.1+ vector (Invitrogen) for expression in HEK293 (human embryonic kidney) cells. The protein alignment was created with MegAlign Pro software v. 17.1.1 (DNASTAR Lasergene) using MUSCLE algorithm with default parameters.

### HcKCR1 and HcCCR expression and purification from *Pichia pastoris*
The plasmids carrying the HcKCR1 and HcCCR expression constructs were linearized with Sac I and delivered into strain SMD1168 by electroporation. Expression and protein purification followed the procedure described[4,41]. A single colony resistant to 0.5 mg/ml zeocin was picked and inoculated into buffered complex glycerol medium, after which the cells were transferred to buffered complex methanol (0.5%) medium supplemented with 5 µM all-trans-retinal (Sigma-Aldrich) and grown at 30 °C with shaking at 230 rpm. After 24 h, the pink-colored cells were harvested by centrifugation at 5000 $g$ for 10 min, and the cell pellets were resuspended in 100 ml ice-cold buffer A (20 mM HEPES, pH 7.4, 150 mM NaCl, 1 mM EDTA, 5% glycerol) and lysed by either French press or bead beater. After centrifugation at low speed (5000 $g$ for 10 min) to remove cell debris, membrane fractions were pelleted at 190,000 $g$ for 1 h using a Ti45 Beckman rotor. The membranes were suspended in Buffer B (350 mM NaCl, 5% glycerol, 20 mM HEPES, pH 7.5) with 1 mM phenylmethylsulfonyl fluoride and solubilized with 1% n-dodecyl-β-D-maltoside (DDM) for 1 h at 4 °C with shaking. Undissolved content was removed after ultracentrifugation using a Ti45 rotor at 110,000 $g$ for 1 h. The supernatant supplemented with 15 mM imidazole was incubated with nickel-nitrilotriacetic acid resin (Qiagen) for 1 h with shaking at 4 °C. The resin was washed in a step-wise manner using 15 mM and 40 mM imidazole in Buffer B supplemented with 0.03% DDM. The protein was eluted with 400 mM imidazole and 0.03% DDM in buffer B. The eluted protein was further purified using a Superdex Increase 10/300 GL column (Cytiva) equilibrated with Buffer B supplemented with 0.03% DDM/Cholesteryl Hemisuccinate (CHS, ratio 10:1). Protein fractions with an A280/A523 absorbance ratio of ~2.0 were pooled, concentrated to ~10 mg/ml using a 50 K MWCO Amicon Ultra Centrifugal Filter (Sigma-Aldrich), flash-frozen in liquid nitrogen and stored at −80 °C until use.

### Absorption and FT Raman spectroscopy
Absorption spectra of the protein samples incubated in the dark for over 30 min were measured with a Cary 4000 spectrophotometer (Varian). Molar protein concentration was calculated using the absorbance value at 523 nm divided by the extinction coefficient 45,000 M$^{-1}$ cm$^{-1}$. FT-Raman spectra were collected on 5 µl of a highly concentrated purified solubilized protein (>30 mg/ml) sealed in a metallic holder, in 180° scattering geometry, with 1064 nm excitation.

FRA106/s accessory to the Bruker IFS66vs spectrometer was used, with the OPUS software, 14,000 scans averaged per sample at a 4 cm$^{-1}$ resolution. The buffer spectrum was collected as a control to ascertain that its lines do not contribute to the spectral regions of interest (ethylenic stretches, HOOPs, and fingerprint vibrations).

## Flash photolysis

Light-induced absorption changes were measured with a laboratory-constructed crossbeam apparatus, which has been described elsewhere[42]. Excitation flashes were from a Minilite II Nd:YAG laser (532 nm, pulsewidth 6 ns, energy 5 mJ; Continuum). Measuring light was from a 250-W incandescent tungsten lamp combined with a McPherson monochromator (model 272, Acton). Absorption changes were detected with a Hamamatsu Photonics photomultiplier tube (model R928) combined with a second monochromator of the same type. Signals were amplified by a low noise current amplifier (model SR445A; Stanford Research Systems) and digitized with a GaGe Octopus digitizer board (model CS8327, DynamicSignals LLC), maximal sampling rate 50 MHz. Logarithmic filtration of the data was performed using the GageCon program[43].

## Reconstitution of proteoliposomes

*Hc*KCR1 in 300 mM NaCl, 20 mM Hepes (pH 7.5), 5% glycerol, 0.05% DDM was reconstituted into liposomes composed of 1-palmitoyl-2-oleoyl-sn-glycero-3-phosphocholine (POPC, Avanti Polar Lipids) similar to reported previously[44]. A film of POPC (initially dissolved in chloroform) was resuspended in 300 mM NaCl, 20 mM Hepes (pH 7.5), 5% glycerol, and then dissolved with 2% DDM (final concentration). *Hc*KCR1 and POPC/DDM were mixed (protein:lipids molar ratio ~1:25) and incubated at 4 °C for 2 h, and subsequently three batches of bio-beads (BioRad) were added to completely remove DDM at 4 °C overnight. The formed proteoliposomes were pelleted by centrifugation at 140,000 $g$ for 30 min and resuspended in 300 mM NaCl, 20 mM Hepes (pH 7.5), 5% glycerol to an absorption of ~0.6 for the visible absorption peak.

## Electron microscopy of *Hc*ChRs in peptidiscs

Purified *Hc*KCR1 or *Hc*CCR was mixed with a 2:1 excess (w/w) of nanodisc scaffolding peptide NSPr (N$_{ter}$-DWLKAFYDKVAEKLKEAAPDW FKAFYDKVAEKFKEAF-C$_{ter}$, purity >80%, A$^+$ peptide Co. Ltd., Shanghai)[10] and diluted to 1/14 of initial concentration before applying to size exclusion chromatography on a Superdex 200 Increase 10/300 GL column in buffer B without glycerol. For initial evaluation, the peak fraction was collected and immediately applied at a concentration of ~0.02 mg/mL onto negatively glow-discharged carbon-coated copper grids (200 mesh, TED PELLA, CAT# 01840-F) for 1 min, and excess liquid was removed by blotting with filter paper and rinsed twice with pure water. Then the grids were negatively stained by freshly prepared 2% uranyl formate and blotted. Images were collected on a Talos L120C transmission electron microscope with 92,000× magnification at the Microscopy Imaging Laboratory at the University of Toronto, as previously described[20]. For cryo-EM, 4 μL of *Hc*KCR1 or *Hc*CCR reconstituted into peptidiscs at 0.35 mg/mL concentration and dark-adapted for >24 h were applied onto homemade holey gold grids[45]. Excess protein was blotted away using a Vitrobot Mark IV set to 277 K, 100% relative humidity, with 2.5 s blot time and blot force 1. Grids were plunge frozen in liquid ethane and stored under liquid nitrogen prior to imaging.

## Cryo-EM image acquisition and data processing

Cryo-EM movies were acquired at 300 kV on a Titan Krios transmission electron microscope equipped with a Falcon 4i detector. Movies consisting of 30 fractions were collected at 75,000× magnification with a pixel size of 1.03 Å and a total exposure of 40e$^-$/Å$^2$. All datasets were collected with a 30° stage tilt. Automated data collection was performed with the EPU (v. 3.3) software package, and a total of 5902 and 12,014 movies were collected for CCR and KCR1 samples, respectively. Movies for the KCR1 dataset were collected over two separate sessions.

Image processing was performed using the CryoSPARC v4.1 software packages[46]. Patch motion correction and CTF estimation were performed, followed by manual curation of exposures to remove poor quality micrographs. A subset of particles was first selected using the blob picker in CryoSPARC to generate 2D classes for reference-based template picking.

With the CCR dataset, 2,674,318 particles were initially picked for 2D classification, and 1,064,296 particles were selected for 3D ab-initio reconstruction. C3 symmetry was applied, and heterogenous refinement followed by non-uniform refinement[47] was performed to obtain a map at 3.03 Å resolution with 296,840 particles. Masks were generated to remove the peptidisc density during refinement, and several iterations of non-uniform and local refinement were performed to obtain a density map at 2.84 Å resolution.

With the KCR1 dataset, 5,692,131 particles were obtained after initial template picking, and 1,689,031 particles were selected after 2D classification. After 3D ab-initio reconstruction and multiple rounds of heterogenous, non-uniform and local refinement, a map with 2.88 Å resolution was obtained with 297,015 particles.

## Model building and refinement

Sharpened density maps prepared in CryoSPARC were used to build models in Coot (v. 0.9.6)[48]. Models for *Hc*KCR1 and *Hc*CCR generated by AlphaFold2[49] were fit to the map in Coot, then the models were further refined in Phenix (v. 1.2)[50]. Structural analysis and figure preparation were performed with UCSF ChimeraX (v. 1.4)[51], PyMOL[52] (v. 2.5.4), APBS[53] (v. 3.4.1), HOLLOW[54] (v. 1.3), and LigPlot+[55] (v. 2.2.4) software.

## H-bond analysis

The structures of *Hc*KCR and *Hc*CCR were aligned along the membrane normal using the PPM web server[56]. H-bond graph computations were performed with the graph-based algorithms Bridge[21] and C-Graphs[22]. The H-bond graph consists of a collection of nodes, which here are H-bonding sidechains, the retinal Schiff base, and water molecules, and edges—the H-bonds between these groups[21]. The conserved H-bond graph of *Hc*KCR and *Hc*CCR consists of the H-bonding groups and H-bonds that are present in both structures; the difference H-bond graph for *Hc*KCR or *Hc*CCR displays all H-bonds identified for each structure according to the criteria used, and marks H-bonds and H-bonding groups conserved in both structures, vs. H-bonds and H-bonding groups unique to either of the structures[22]. To facilitate comparison of the difference graphs computed for *Hc*KCR and *Hc*CCR, the H-bond graphs were projected along the membrane normal, which corresponds to the $z$-coordinate of the Cartesian coordinate system. The membrane ($x$-$y$) plane projection of the H-bonding groups is done using Scikit-learn[57] in C-Graphs[22].

H-bonds were computed using as criterion a distance of ≤4.0 Å between the H-bond donor and acceptor heteroatoms. This H-bond distance, which is longer than the 3.5 Å distance typically used, was chosen to include in the graphs the weaker interactions of Met sidechains. To distinguish between H-bonds that satisfy the stricter 3.5 Å distance criterion and those found only with the 4 Å criterion, the former are marked with thicker edges.

## Pore analysis

Analyses of the conduction pathway pore were done using HOLLOW[54] (v. 1.3) using a grid spacing of 0.5 Å and a surface probe of 1.4 Å. The surface visualized in the final figures was created by selecting manually "dummy waters" generated by HOLLOW. Electrostatic potentials were calculated using APBS[53] (v. 3.4.1). The conduction pathway was also

analyzed using the program CAVER Analyst 2.0 Beta[58] with a probe radius of 0.9 Å.

### HEK293 cell culture and transfection
No cell lines from the list of known misidentified cell lines maintained by the International Cell Line Authentication Committee were used in this study. HEK293 cells were obtained from the American Type Culture Collection (ATCC; catalog #CRL-1573) and grown as described earlier[4]. The cells were plated on 2-cm diameter plastic dishes 48–72 h before experiments, grown for 24 h and transfected with 10 μl of ScreenFectA transfection reagent (Waco Chemicals USA, Richmond) using 3 μg DNA per dish. All-*trans*-retinal (Sigma) was added immediately after transfection at the final concentration of 5 μM.

### Whole-cell patch clamp recording
Whole-cell patch clamp recordings from transfected HEK293 cells were performed in voltage clamp mode with an Axopatch 200B amplifier (Molecular Devices) at room temperature (25 °C). The signals were digitized with a Digidata 1440A using pCLAMP ClampEx 10.7 software (both from Molecular Devices). Patch pipettes with resistances of 1.5–2.5 MΩ were fabricated from borosilicate glass. The pipette solution contained (in mM): KCl 130, MgCl$_2$ 2, HEPES 10 pH 7.4, and the bath solution contained (in mM): NaCl 130, CaCl$_2$ 2, MgCl$_2$ 2, glucose 10, HEPES 10 pH 7.4. A 4 M salt bridge was used in all experiments. All holding voltages were corrected for liquid junction potentials (LJP) calculated using the ClampEx built-in LJP calculator. Light pulses were provided by a Polychrome IV light source (T.I.L.L. Photonics GMBH) in combination with a mechanical shutter (Uniblitz Model LS6, Vincent Associates, Rochester; half-opening time 0.5 ms). Maximal quantum density at the focal plane of the 40× objective lens was ~7 mW mm$^{-2}$ at 540 nm. The $V_{rev}$ values were determined by recording a series of photocurrents in response to incrementally varied voltage steps, measuring their amplitudes using pCLAMP ClampFit 10.7 software (Molecular Devices) and plotting the current–voltage dependencies (*IV* curves) using Origin Pro 2019 software (OriginLab corporation). The action spectra were constructed by calculation of the initial slope of photocurrent in the linear range of the dependence on the quantum density, corrected for the quantum density measured at each wavelength and normalized to the maximal value.

### Statistics and reproducibility
Identical batches of HEK293 cell culture were randomly assigned for transfection with each tested construct. Individual transfected HEK293 cells were selected for patching by inspecting their tag fluorescence; non-fluorescent cells and cells in which no GΩ seal was established or lost during recording were automatically excluded from measurements. One photocurrent trace was recorded from each cell, and traces recorded from different cells transfected with the same construct were considered biological replicates (reported as *n* values). These values indicate how often the experiments were performed independently. Descriptive statistics was calculated by Origin Pro 2019 software. The data are presented as mean ± sem values, as indicated in the figure captions; the data from individual cells are also shown when appropriate. No statistical methods were used to pre-determine sample sizes but our sample sizes are similar to those reported in previous publications[3,8,10]. Normal distribution of the data was tested with the Shapiro-Wilk test; if passed, one-way ANOVA with the Tukey test for means comparison was used; if failed, the non-parametric Mann-Whitney test was used as implemented in Origin software (OriginLab Corporation).

### Reporting summary
Further information on research design is available in the Nature Portfolio Reporting Summary linked to this article.

## Data availability
Source data are provided with this paper. The cryo-EM maps have been deposited in the Electron Microscopy Data Bank (EMDB) under accession codes EMD-40062 (*Hc*KCR1) and EMD-40063 (*Hc*CCR). The coordinates have been deposited in the RCSB Protein Data Bank (PDB) under accession codes 8GI8 (*Hc*KCR1) and 8GI9 (*Hc*CCR). Source data are provided with this paper.

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

## Acknowledgements

We thank Maryam Khoshouei, Emil F. Pai, and Benjamin Kaupp for helpful discussions. We thank Samir Benlekbir and Zhijie Li of the Toronto High-Resolution High Throughput cryo-EM facility, supported by the Canada Foundation for Innovation and Ontario Research Fund, for data acquisition and advice on preparation of cryo-EM gold grids. This work was supported by the National Institutes of Health grants R35GM140838 and U01NS118288 (J.L.S.) and R01GM143418 (L.Z.); Robert A. Welch Foundation Endowed Chair AU-0009 (J.L.S.); Natural Sciences and Engineering Research Council of Canada Discovery Grants RGPIN-2018-04397 (L.S.B.) and RGPIN-2017-06862 (O.P.E.) and the Canadian Institutes of Health Research (CIHR) Operating Grant PJT-159464 (O.P.E.).

## Author contributions

T.M., K.K., H.L., and E.G.G. contributed equally to this work. T.M. and K.K. reconstituted HcChRs into peptidiscs, performed cryo-EM sample preparation, data acquisition and processing, structure modeling and refinement. H.L., L.Z., Y.W., and A.A.K. cloned, expressed and purified

HcChRs. E.G.G. performed electrophysiology experiments. H.L., E.G.G., and L.Z. helped with structure interpretation. O.A.S., A.A.K., and L.S.B. performed spectroscopic studies, E.B. and A.N.B. did the hydrogen bond analysis, J.L.S. and O.P.E. supervised the project. T.M., K.K., E.G.G., L.S.B., and E.B. prepared the figures, T.M., E.G.G., and O.P.E. prepared the manuscript with contributions from all authors.

## Competing interests

The authors declare no competing interests.
