## [Peer Review File · Nature Communications]

Structures of channelrhodopsin paralogs in peptidiscs explain their contrasting K⁺ and Na⁺ selectivitiesREVIEWER COMMENTS

Reviewer #1 (Remarks to the Author):

The manuscript by Morizumi et al. entitled “Structures of channelrhodopsin paralogs in peptidiscs explain their contrasting K⁺ and Na⁺ selectivities” is a well-written description of a study in which the ion selectivity in K⁺ and Na⁺ channelrhodopsins is explored by cryo-electron microscopy. The two target proteins are both from *Hyphochytrium catenoides*: potassium-selective kalium channelrhodopsin 1 HcKCR1 and the sodium-selective cation channelrhodopsin HcCCR. The dark state structures are very similar but differ at two sites flanking the retinal binding site and counterion that function in channel gating: an intracellular segment containing several critical residues and a cluster of aromatic residues on the extracellular side. Accompanying mutagenesis and patch clamp studies support the role of these two sites as critical determinants of ion selectivity in the two channels.

The study is of interest for several reasons. Of primary concern is the potential to improve the kalium channel for use as an optogenetic tool for silencing of mammalian neurons. Another point of interest is the use of peptidiscs in preparation of these integral membrane proteins for cryo-electron microscopy. The appearance of this manuscript is likely to initiate a significant shift from Nanodiscs to peptidiscs as the system of choice for solubilization of these and other membrane proteins in elucidation of structure and function. Overall, the study presented in this manuscript is timely and will appeal to a broad readership.

Reviewer #2 (Remarks to the Author):

The manuscript “Structures of channelrhodopsin paralogs in peptidiscs explain their contrasting K⁺ and Na⁺ selectivities” by Morizumi et al. describes the results of cryoEM structural studies of two channelrhodopsins: kalium channelrhodopsin (HcKCR1) and sodium channelrhodopsin (HcCCR) from *Hyphochytrium catenoides*. In contrast to several conventional rhodopsin cation channels permeable for different ions, including divalent ones, HcKCR1 and HcCCR channels are highly selective. High selectivity to K⁺ and Na⁺ is a big advantage if one thinks about optogenetic applications. The structures of both proteins solved by the authors of the manuscript and taken together provide important information on the mechanisms of rhodopsin ion channels, in particular, on the mechanisms of ion selectivity. The mechanisms of channelrhodopsins are still a great challenge in the field under discussion and, as it is also widely considered, a bottleneck for the development of advanced optogenetic tools.

The experimental work is done accurately, the analysis of the results is quite deep and the conclusions are mostly adequate. Therefore, despite the fact that in order to disclose the true mechanisms it is

necessary to solve the structures of the proteins with the open channel, I consider that this work is important and can be potentially published in Nature Communications.

Nevertheless, a major revision is necessary to improve the manuscript. First of all, I agree that comparison of two closed proteins is very informative to understand the preference of the proteins to K⁺ or Na⁺ permeability and is methodologically correct. However, comparison with a nonselective channel rhodopsin is of high importance as well. It is necessary. An ideal reference would be classical channel rhodopsin 2 (ChR2) where vast functional and structural information is available. Second, the authors know that “The closest relatives of KCRs among other ChRs are “bacteriorhodopsin (BR)-like cation channelrhodopsins” (BCCRs) from cryptophyte algae, none of which exhibits K⁺ selectivity.” It would be logical to include a BCCR (for example, ChRmine) into the analysis. Third, the presented in the manuscript about 3 Å resolution cryoEM structures are not sufficient to reliably identify all water molecules and conformations of some of amino acids. It is known that in the case of some rhodopsins such problems made difficult interpretation of the structures in terms of the mechanisms. The authors use modelling to support the experimental data. I would suggest discussing the accuracy of the data and possible problems of the interpretation of such models.

Some minor remarks are below.

Abstract

- First sentence of the abstract is a little bit long.
- It would be informative to specify “critical residues” of “two distinct sites.”

Introduction

- It would be useful to compare the both proteins with ChR2

Results

- It is useful to mention in the beginning of the main text at what pH the structures were solved. Protonation/deprotonation of functionally important proton donor residues depend on pH and therefore may influence the structures.
- The authors successfully applied peptidiscs in cryoEM structural biology of two rhodopsins. It might be of a wide interest. It is known that lipid or lipid-mimicking environment of membrane proteins may, sometimes strongly, influence protein function. Could you provide evidence that pepti belt around the rhodopsins did not change their function and structure. Could you compare the photocycles of the proteins measured with the proteins reconstituted in lipid vesicles and in peptidiscs.
- Page 4. Fourth paragraph. Fixation of Na⁺ even at higher resolution is not trivial. Could you show the densities around the sodium ion and the corresponding coordination bonds?
- Page 5. Third paragraph. To avoid confusion with the “dark-adapted state”, please, define the “dark state”.
- Page 7. Third paragraph. It is useful for a wide reader if you replace “...defines the absorption properties...” with “...defines light absorption properties.”

- Page 8. closing sentences. It is important to specify, if possible, whether hydrated or dehydrated ions pass the channels.
- Figure 3. In accordance with Figure 3b there is a continuous (?) pore extending from the RSB to the extracellular bulk. The authors present the structure of the ground state of HcCCR. It is somehow unusual. Is it not an artifact of the presentation? Did you observe a small dark current?
- Figure 5. One cannot clearly see the famous pentamer of H-bonds. Could you show the pentamer. It can be useful for comparison.

Reviewer #3 (Remarks to the Author):

I have reviewed the manuscript entitled "Structures of channelrhodopsin paralogs in peptidiscs explain their contrasting K⁺ and Na⁺ selectivities" Overall, the manuscript presents an interesting study on the mechanisms of K⁺ selectivity in microbial rhodopsins, with a particular focus on HcKCR1 and HcCCR. The authors used cryo-electron microscopy to investigate the roles of specific residues in the proteins' structures and functions. The results are insightful and provide a deeper understanding of the mechanisms underlying K⁺ selectivity in these proteins. However, there are several issues with the manuscript that need to be addressed before it can be accepted for publication.

1. On page 5, "the side chain of Asp116 is predicted to be deprotonated at neutral pH (pKa approx. 6 and 5, respectively, estimated by PROPKA software²³)... ease of breaking the salt bridge to Arg244". If the pKa values are 5 to 6, Asp116 and Arg244 do not form a salt bridge (pKa is too high). Without distance information, it is difficult to determine whether or not they form an H-bond (including salt bridge). Therefore, I suggest that the authors include distance information for all hydrogen bonds in the main text and figures (e.g. Figure 4), including this salt bridge.

2. On page 6, the authors mention that "The H-bond connection from Tyr106 to His225 is stronger in HcCCR than in HcKCR1." However, the authors should be careful when using the term "strong hydrogen bonds" (or don't use unless specified). They may find it helpful to refer to the articles DOI: 10.1146/annurev.physchem.48.1.511 and DOI: 10.1002/prot.20096 for a better understanding of the term's appropriate usage.

3. On page 7, the authors refer to "the protonated water cluster in BR". While I understand their intended meaning, I believe that this phrase may not be an appropriate choice, since the presence of H₃O⁺ would require a specific protein environment that is not present in BR (although some have proposed its existence). See DOI: 10.1002/ange.201705512

4. In Figure 3, the authors attempt to depict the network of cavities and cation-conducting pathways, but the point they are trying to make is unclear. It would be helpful if the authors analyzed the cavity space using Caver to provide a more comprehensive analysis. Additionally, it should be noted that the conducting pathway may not necessarily be open in the original structure, as demonstrated in anion channel rhodopsins (see Figure 6 in DOI: <https://doi.org/10.7554/eLife.72264>). This point should be taken into consideration when discussing the conducting pathway.

We thank all three Reviewers for their careful reading of our manuscript and their very helpful suggestions to improve it. To address the Reviewers' comments, we have modified in the revised manuscript figure 5, added 8 new supplementary figures showing new data and/or analyses, added the corresponding methods, and added a Supplementary Discussion. Following the guidelines to authors, we also removed the footnote on page 8. Below we provide our point-by-point response (in blue) to the Reviewers' comments (in black).

Reviewer #1 (Remarks to the Author):

The manuscript by Morizumi et al. entitled "Structures of channelrhodopsin paralogs in peptidiscs explain their contrasting K⁺ and Na⁺ selectivities" is a well-written description of a study in which the ion selectivity in K⁺ and Na⁺ channelrhodopsins is explored by cryo-electron microscopy. The two target proteins are both from *Hyphochytrium catenoides*: potassium-selective kalium channelrhodopsin 1 *HcKCR1* and the sodium-selective cation channelrhodopsin *HcCCR*. The dark state structures are very similar but differ at two sites flanking the retinal binding site and counterion that function in channel gating: an intracellular segment containing several critical residues and a cluster of aromatic residues on the extracellular side. Accompanying mutagenesis and patch clamp studies support the role of these two sites as critical determinants of ion selectivity in the two channels.

The study is of interest for several reasons. Of primary concern is the potential to improve the kalium channel for use as an optogenetic tool for silencing of mammalian neurons. Another point of interest is the use of peptidiscs in preparation of these integral membrane proteins for cryo-electron microscopy. The appearance of this manuscript is likely to initiate a significant shift from Nanodiscs to peptidiscs as the system of choice for solubilization of these and other membrane proteins in elucidation of structure and function. Overall, the study presented in this manuscript is timely and will appeal to a broad readership.

We thank the Reviewer for his/her positive evaluation of our study.

Reviewer #2 (Remarks to the Author):

The manuscript "Structures of channelrhodopsin paralogs in peptidiscs explain their contrasting K⁺ and Na⁺ selectivities" by Morizumi et al. describes the results of cryoEM structural studies of two channelrhodopsins: kalium channelrhodopsin (*HcKCR1*) and sodium channelrhodopsin (*HcCCR*) from *Hyphochytrium catenoides*. In contrast to several conventional rhodopsin cation channels permeable for different ions, including divalent ones, *HcKCR1* and *HcCCR* channels are highly selective. High selectivity to K⁺ and Na⁺ is a big advantage if one thinks about optogenetic applications. The structures of both proteins solved by the authors of the manuscript and taken together provide important information on the mechanisms of rhodopsin ion channels, in particular, on the mechanisms of ion selectivity. The mechanisms of channelrhodopsins are still a great challenge in the field under discussion and, as it is also widely considered, a bottleneck for the development of advanced optogenetic tools.

The experimental work is done accurately, the analysis of the results is quite deep and the conclusions are mostly adequate. Therefore, despite the fact that in order to disclose the true mechanisms it is necessary to solve the structures of the proteins with the open channel, I consider that this work is important and can be potentially published in Nature Communications.

Nevertheless, a major revision is necessary to improve the manuscript. First of all, I agree that comparison of two closed proteins is very informative to understand the preference of the proteins to K⁺ or Na⁺ permeability and is methodologically correct. However, comparison with a nonselective channel rhodopsin is of high importance as well. It is necessary. An ideal reference would be classical channel rhodopsin 2 (ChR2) where vast functional and structural information is available.

In the revision we have added new supplementary figures (Supplementary Figures 8-10) showing comparison of *HcKCR1* with ChRmine and ChR2, and described it in the Supplementary Discussion "Comparison of *HcKCR1* and *HcCCR* with other CCR structures".

We added the following sentence at the end of the third paragraph on page 4:

“While we focus here on *HcKCR1* and *HcCCR*, we provide in the supplement a comparison of *HcKCR1* with ChR2 and ChRmine (Supplementary Figs. 8-10, Supplementary Discussion).”

Please, however, note that referring to ChR2 as “non-selective” and to *HcKCR1*/*HcCCR* as “selective” may be misleading, as all these ChRs are permeable to H⁺, Na⁺ and K⁺, although their relative permeabilities differ.

Second, the authors know that “The closest relatives of KCRs among other ChRs are “bacteriorhodopsin (BR)-like cation channelrhodopsins” (BCCRs) from cryptophyte algae, none of which exhibits K⁺ selectivity.” It would be logical to include a BCCR (for example, ChRmine) into the analysis.

In the revision we have added new supplementary figures (Supplementary Figures 8-10) showing comparison of *HcKCR1* with ChRmine and ChR2, and described it in the Supplementary Discussion "Comparison of *HcKCR1* and *HcCCR* with other CCR structures".

Third, the presented in the manuscript about 3 Å resolution cryoEM structures are not sufficient to reliably identify all water molecules and conformations of some of amino acids. It is known that in the case of some rhodopsins such problems made difficult interpretation of the structures in terms of the mechanisms. The authors use modelling to support the experimental data. I would suggest discussing the accuracy of the data and possible problems of the interpretation of such models.

We would like to note that the local resolution in the transmembrane domain where we modeled water molecules is ~ 2.5 Å, which allows us to identify the waters and side chain conformations. The H-bond computations did not involve any modeling; we followed published procedures developed by co-authors Bertalan and Bondar (Frontiers in Chemistry, 2023, DOI=10.3389/fchem.2022.1075648), which works nicely for microbial rhodopsins in that resolution range (see Fig. 1 of the Bertalan and Bondar paper).

Some minor remarks are below.

Abstract

- First sentence of the abstract is a little bit long.

In the revised manuscript we have split the first sentence of the abstract as follows:

“Kalium channelrhodopsin 1 from *Hyphochytrium catenoides* (*HcKCR1*) is a light-gated channel used for optogenetic silencing of mammalian neurons. It selects K^+ over Na^+ in the absence of the canonical tetrameric K^+ selectivity filter found universally in voltage- and ligand-gated channels.”

- It would be informative to specify “critical residues” of “two distinct sites.”

In the revised manuscript we have specified the critical residues (highlighted in bold) as follows:

“Together with structure-guided mutagenesis, we found that K^+ versus Na^+ selectivity is determined at two distinct sites on the putative ion conduction pathway: in a patch of critical residues in the intracellular segment (**Leu69, Ile73 and Asp116**) and within a cluster of aromatic residues in the extracellular segment (**primarily, Trp102 and Tyr222**).”

Due to this 8 words extra information, the abstract increases to 158 words. The editor may decide whether this addition is tolerable.

Introduction

- It would be useful to compare the both proteins with ChR2

In the revision we have added new supplementary figures (Supplementary Figures 8-10) showing comparison of *HcKCR1* with ChRmine and ChR2, and described it in the Supplementary Discussion "Comparison of *HcKCR1* and *HcCCR* with other CCR structures".

Results

- It is useful to mention in the beginning of the main text at what pH the structures were solved. Protonation/deprotonation of functionally important proton donor residues depend on pH and therefore may influence the structures.

We added this information "pH 7.5" to the second sentence of the results:

“We reconstituted purified *HcKCR1* as well as *HcCCR* at pH 7.5 into peptidiscs and imaged them by cryo-EM.”

- The authors successfully applied peptidiscs in cryoEM structural biology of two rhodopsins. It might be of a wide interest. It is known that lipid or lipid-mimicking environment of membrane proteins may, sometimes strongly, influence protein function. Could you provide evidence that pepti belt around the rhodopsins did not change their function and structure. Could you compare the photocycles of the proteins measured with the proteins reconstituted in lipid vesicles and in peptidiscs.

As suggested by the reviewer we measured the photochemical conversions of *HcKCR1* reconstituted in liposomes and in peptidiscs and added the data as Supplementary Fig. 3. We added information on reconstitution of *HcKCR1* into liposomes to the methods and describe the results by modifying the first paragraph of the results:

“The peptidisc environment had little effect on the function as determined by measuring the photochemical conversions of *HcKCR1* reconstituted in liposomes and peptidiscs (Supplementary Fig. 3). The density maps of the ChR trimers obtained from cryo-EM imaging are shown for *HcKCR1* in Figure 1.”

- Page 4. Fourth paragraph. Fixation of Na⁺ even at higher resolution is not trivial. Could you show the densities around the sodium ion and the corresponding coordination bonds?

In the text on page 4 we wrote in the original manuscript: “Small spherical densities within the cavities were interpreted as water molecules except those that show electrostatic interactions with aromatic systems and lack hydrogen bonding (H-bonding), which were interpreted as Na⁺ buffer component (Supplementary Fig. 11).”

We now added "Supplementary Fig. 11" to the sentence to illustrate this.

- Page 5. Third paragraph. To avoid confusion with the “dark-adapted state”, please, define the “dark state”.

We replaced in this paragraph:

“Comparison of the dark-state *HcKCR1* and *HcCCR* structures suggests...” with

“Comparison of the **closed-state (dark)** *HcKCR1* and *HcCCR* structures suggests...”

to make it uniform with page 4, paragraph 4.

- Page 7. Third paragraph. It is useful for a wide reader if you replace “...defines the absorption properties...” with “...defines light absorption properties.”

The requested replacement has been made in the revision.

- Page 8. closing sentences. It is important to specify, if possible, whether hydrated or dehydrated ions pass the channels.

As mentioned in the manuscript, the structures we obtained are the structures of the closed channel. They do not provide sufficient information to decide whether the cations pass the channel in the hydrated or dehydrated state, so we think that any statement about this would be too speculative at this stage of the research.

- Figure 3. In accordance with Figure 3b there is a continuous (?) pore extending from the RSB to the extracellular bulk. The authors present the structure of the ground state of *HcCCR*. It is somehow unusual. Is it not an artifact of the presentation? Did you observe a small dark current?

The presence of a vestibule leading from the extracellular space to the center of the molecule is not unusual among channelrhodopsins, as, e.g., the structure of the hybrid channelrhodopsin C1C2 (PDB ID: 3UG9) shows. In the revision, we have added the new Supplementary Fig. 18 showing the results of CAVER analysis of intramolecular tunnels in *HcKCR1*, *HcCCR* and C1C2. Both *HcCCR* and C1C2 show the

tunnel detected with the probe radius 0.9 angstrom leading from the outside to the center of the molecule, whereas no such tunnel could be detected in *HcKCR1*. Please note that the presence of such a tunnel in the extracellular segment does not mean that *HcCCR* and *C1C2* pass cations in the dark, as these tunnels do not extend to the intracellular aqueous phase.

- Figure 5. One cannot clearly see the famous pentamer of H-bonds. Could you show the pentamer. It can be useful for comparison.

We thank the reviewer for pointing this out. We modified Figure 5 in the revision and changed the orientation of the molecules to show the water-mediated pentamer of H-bonds in BR. No such pentamer is found in *HcKCR1* or *HcCCR* because the Schiff base forms an H-bond to D229 instead of to a water molecule in BR. We added the following sentence to the last paragraph on page 6:

“In contrast, the Schiff base of BR is connected via a water molecule to the H-bonding network (Fig. 5c).”

Reviewer #3 (Remarks to the Author):

I have reviewed the manuscript entitled "Structures of channelrhodopsin paralogs in peptidiscs explain their contrasting K⁺ and Na⁺ selectivities" Overall, the manuscript presents an interesting study on the mechanisms of K⁺ selectivity in microbial rhodopsins, with a particular focus on *HcKCR1* and *HcCCR*. The authors used cryo-electron microscopy to investigate the roles of specific residues in the proteins' structures and functions. The results are insightful and provide a deeper understanding of the mechanisms underlying K⁺ selectivity in these proteins. However, there are several issues with the manuscript that need to be addressed before it can be accepted for publication.

1. On page 5, “the side chain of Asp116 is predicted to be deprotonated at neutral pH (pK_a approx. 6 and 5, respectively, estimated by PROPKA software²³)... ease of breaking the salt bridge to Arg244”. If the pK_a values are 5 to 6, Asp116 and Arg244 do not form a salt bridge (pK_a is too high). Without distance information, it is difficult to determine whether or not they form an H-bond (including salt bridge). Therefore, I suggest that the authors include distance information for all hydrogen bonds in the main text and figures (e.g. Figure 4), including this salt bridge.

We thank the reviewer for this suggestion. When we did our H-bond network analysis (Fig. 3), we also determined the distances between the atoms. We tried to add these numbers to the figure, but the figure then appears to be too crowded. We therefore added Supplementary Figs. 12 and 13 analogous to Fig. 3c,d which provides all distances. Further, we removed the pK_a analysis and modified the text from:

“In both HcKCR1 and HcCCR, the side chain of Asp116 is predicted to be deprotonated at neutral pH (pK_a approx. 6 and 5, respectively, estimated by PROPKA software²³) likely due to the interaction with Arg244 and an environment of fewer Leu and Phe residues compared with BR. The density of Asp116 is not well-defined arguing for some flexibility and ease of breaking the salt bridge to Arg244 to open up the constriction for cation passage.”

to:

“In both *HcKCR1* and *HcCCR*, the side chains of Asp116 and Arg244 are interacting. The distance between these side chains is 3.4 Å for *HcKCR1* and 3.0 Å for *HcCCR*, respectively (Supplementary Figs. 12 and 13). The density of Asp116 is not well-defined, arguing for flexibility and ease of breaking its interaction with Arg244 to open up the constriction for cation passage.”

2. On page 6, the authors mention that "The H-bond connection from Tyr106 to His225 is stronger in *HcCCR* than in *HcKCR1*." However, the authors should be careful when using the term "strong hydrogen bonds" (or don't use unless specified). They may find it helpful to refer to the articles DOI: 10.1146/annurev.physchem.48.1.511 and DOI: 10.1002/prot.20096 for a better understanding of the term's appropriate usage.

To avoid any confusion, the wording 'stronger' or 'strength' are no longer used in reference to H-bonds discussed in the manuscript. Instead, we state the length of H-bonds and changed the wording to "Tyr106 and His225 are within 3.6 Å distance in *HcKCR1*, as compared to 3.4 Å in *HcCCR* (Supplementary Figs. 12 and 13)."

3. On page 7, the authors refer to "the protonated water cluster in BR". While I understand their intended meaning, I believe that this phrase may not be an appropriate choice, since the presence of H₃O⁺ would require a specific protein environment that is not present in BR (although some have proposed its existence). See DOI: 10.1002/ange.201705512

We thank the reviewer for raising this point and changed the wording from:

Thr205 that is linked to the protonated water cluster in BR, is conserved in HcCCR (Thr222) but replaced with Tyr in HcKCR1.

to:

“Thr205 in BR is analogous to Thr222 in *HcCCR* and replaced with Tyr222 in *HcKCR1*. While the side chains of all three residues form an H-bond to a water molecule, Tyr222 in *HcKCR1* forms additional H-bonds to Trp102 and Gln218.”

4. In Figure 3, the authors attempt to depict the network of cavities and cation-conducting pathways, but the point they are trying to make is unclear. It would be helpful if the authors analyzed the cavity space using Caver to provide a more comprehensive analysis. Additionally, it should be noted that the conducting pathway may not necessarily be open in the original structure, as demonstrated in anion channel rhodopsins (see Figure 6 in DOI: <https://doi.org/10.7554/eLife.72264>). This point should be taken into consideration when discussing the conducting pathway.

As noted in our manuscript, the structures we obtained are structures of the closed channels, as we imaged dark-adapted protein samples. Nevertheless, it is reasonable to expect that the conduction pathway that appears upon illumination is formed by expansion and merging of the cavities observed in the dark state. Therefore, comparison of the size and distribution of such cavities is important, although no direct prediction of the pore size in the open structure can be made from it. Upon the Reviewer's request, we have carried out CAVER analysis of intramolecular tunnels in *HcKCR1* and *HcCCR* and added the new Supplementary Fig. 18 in the revision. For comparison, we have also included C1C2 in our analysis.

We added the following sentence to the result section with subheading "Extracellular segment with a cluster of aromatic residues" where we analyze the extracellular pore:

"Analysis of the *HcChR* structures with the program CAVER confirmed an extracellular tunnel for *HcCCR*, similar to that of the hybrid channelrhodopsin C1C2³¹, but lack of a tunnel for *HcKCR1* (Supplementary Fig. 18)."

REVIEWERS' COMMENTS

Reviewer #2 (Remarks to the Author):

The authors replied properly to my comments

They also made the corresponding changes/corrections in the text and Figures

I would support publication of the manuscript in Nature Communications

Reviewer #3 (Remarks to the Author):

The authors have made commendable efforts in addressing the reviewer's concerns, and the reviewer appreciates the revisions they have made based on my comments. However, there are still some areas that could benefit from further improvement, specifically addressing the remaining suggestions made by the reviewer. Taking these suggestions into consideration will help strengthen the manuscript and enhance its overall quality.

Response to reviewer

Reviewer #3 (Remarks to the Author):

The authors have made commendable efforts in addressing the reviewer's concerns, and the reviewer appreciates the revisions they have made based on my comments. However, there are still some areas that could benefit from further improvement, specifically addressing the remaining suggestions made by the reviewer. Taking these suggestions into consideration will help strengthen the manuscript and enhance its overall quality.

We thank the reviewer. We think that the remaining suggestions may address discussion of the channel in the extracellular segment. We therefore made that part clearer in the paragraph "Extracellular segment with a cluster of aromatic residues" on page 7 and added one more reference (new Ref. 31, Tsujimura et al 2021). This slightly extended part of the paragraph now reads:

In *HcKCR1* Tyr222 forms H-bonds to Trp102 and Gln218. As a result, the extracellular channel opening is interrupted with the bulky aromatic Tyr side chain in *HcKCR1* but continues deeper into the molecule in *HcCCR*, creating a key determinant for K⁺ selectivity. Analysis of the closed-state (dark) *HcChR* structures with the program CAVER confirmed an extracellular tunnel for *HcCCR* where cavities are merged, similar to C1C2 ChR³⁰, but lack of an extracellular tunnel for *HcKCR1* where cavities are separated and the H-bonding network of Tyr222 is expected to be altered by retinal isomerization (Supplementary Fig. 18). Similarly, alteration of a H-bonding network was postulated for fast channel closing in *Guillardia theta* anion channelrhodopsin 1 (*GtACR1*)³¹.